# Genome-Wide Analyses Identifies Known and New Markers Responsible of Chicken Plumage Color

**DOI:** 10.3390/ani10030493

**Published:** 2020-03-15

**Authors:** Salvatore Mastrangelo, Filippo Cendron, Gianluca Sottile, Giovanni Niero, Baldassare Portolano, Filippo Biscarini, Martino Cassandro

**Affiliations:** 1Dipartimento Scienze Agrarie, Alimentari e Forestali, University of Palermo, 90128 Palermo, Italy; salvatore.mastrangelo@unipa.it (S.M.); baldassare.portolano@unipa.it (B.P.); 2Dipartimento di Agronomia Animali Alimenti Risorse Naturali e Ambiente, University of Padova, 35020 Legnaro, Italy; g.niero@unipd.it (G.N.); martino.cassandro@unipd.it (M.C.); 3Dipartimento Scienze Economiche, Aziendali e Statistiche, University of Palermo, 90128 Palermo, Italy; gianluca.sottile@unipa.it; 4CNR-IBBA, 20133 Milano, Italy; filippo.biscarini@gmail.com

**Keywords:** local chicken populations, genome-wide analyses, SNP, plumage color, candidate genes

## Abstract

**Simple Summary:**

In order to assess sources of variation related to Polverara breed plumage color (black vs. white), we carried out genome-wide analyses to identify the genomic regions involved in this trait. The present work has revealed new candidate genes involved in the phenotypic variability in local chicken populations. These results also contribute insights into the genetic basis for plumage color in poultry, and confirm the great complexity of the mechanisms that control this trait.

**Abstract:**

Through the development of the high-throughput genotyping arrays, molecular markers and genes related to phenotypic traits have been identified in livestock species. In poultry, plumage color is an important qualitative trait that can be used as phenotypic marker for breed identification. In order to assess sources of genetic variation related to the Polverara chicken breed plumage colour (black vs. white), we carried out a genome-wide association study (GWAS) and a genome-wide fixation index (*F*_ST_) scan to uncover the genomic regions involved. A total of 37 animals (17 white and 20 black) were genotyped with the Affymetrix 600 K Chicken single nucleotide polymorphism (SNP) Array. The combination of results from GWAS and *F*_ST_ revealed a total of 40 significant markers distributed on GGA 01, 03, 08, 12 and 21, and located within or near known genes. In addition to the well-known *TYR*, other candidate genes have been identified in this study, such as *GRM5*, *RAB38* and *NOTCH2.* All these genes could explain the difference between the two Polverara breeds. Therefore, this study provides the basis for further investigation of the genetic mechanisms involved in plumage color in chicken.

## 1. Introduction

Over the last century, erosion of livestock genetic resources has been observed as a result of the massive replacement of low-productivity local breeds with highly productive ones. These local breeds are nonetheless an important reservoir of genetic diversity, each with specific characteristics. Local animal genetic resources might indeed be characterized by specific heritable phenotypes potentially relevant for current or future use in breeding programs [1]. Several studies showed that local populations can be useful for the investigation of the genetic factors underlying those unique phenotypes related to their diversity [2,3,4].

In Italy, there are numerous known local chicken breeds whose overall conservation status is nevertheless critical; with the abandoning of farming in marginal areas and the advent of industrial-scale chicken breeding, highly specialized chicken lines have replaced the less competitive local breeds [5]. The interest in the conservation of Italian local chicken breeds emerged from an in situ marker-assisted conservation scheme, that involved seven breeds reared in region Veneto: Ermellinata di Rovigo, Pepoi, Robusta Lionata, Robusta Maculata, Millefiori di Lonigo, Padovana and Polverara. The latter is an ancient dual-purpose chicken breed, named after a small town south of Padua. The early history of the Polverara breed is unclear, but it is believed to be the result of a cross between Padovana and other local Veneto chicken populations [6]. The Polverara is a medium-sized chicken with a feathery crest, that erects over the head without covering the eyes. Two different monochrome plumage colors are officially recognized for the Polverara breed, black and white, resulting in two populations: Polverara White (PW) and Polverara Black (PB). Additional Polverara color-varieties may result from crossbreeding between the breed with other local fowls, but they are not standardized. PW and PB are reared separately, and cross-breeding is not commonly practiced, or at least not recorded. Evidence from previous studies shows close genetic relationships between the two Polverara populations [5,7].

As a consequence of their features (phenotypic differentiation and common genetic background), these two populations provide an interesting model to study the genomic regions underpinning their phenotypic diversity, in particular the plumage color.

Alongside the advance of high-throughput genotyping arrays, molecular markers and genes associated to phenotypic traits or diseases in chickens have been identified through genome-wide approaches [8,9,10]. In this study, we carried out a genome-wide association study (GWAS) and a genome-wide fixation index (*F*_ST_) scan to identify genomic regions that may explain the phenotypic differences observed between PW and PB.

## 2. Materials and Methods 

### 2.1. DNA Samples, Genotyping and Quality Control

The collection of blood samples was conducted as part of routine health screening by qualified veterinarians following guidelines established by Institutional Animal Care and Use Committee (IACUC). 

Blood samples were collected from ulnar veins from 37 unrelated animals belonging to the Polverara White (PW) (n = 17) and Polverara Black (PB) (n = 20) chicken breeds (Figure 1). The animals were randomly selected from three different conservation centers located in different areas of Veneto. DNA samples were genotyped using Affymetrix Axiom 600 K Chicken Genotyping Array containing 580,954 single nucleotide polymorphisms (SNPs). The Gallus_gallus-5.0 chicken genome assembly was used in this study as a reference. Only markers located on chromosomes 1 to 28 were used. 

Quality control procedures were performed for the genotype data using PLINK 1.9 [11]. The following filtering parameters were adopted: (i) SNPs with call rate <95%, (ii) minor allele frequency <5% and (iii) animals with more than 10% of missing genotypes were removed.

### 2.2. Genome-Wide Analyses

We performed a genome-wide association study (GWAS) using the univariate case-control model (PW vs. PB) implemented in the *snpassoc* R package [12], specifically the log-additive genetic model. We used Bonferroni correction to determine the genome-wide significance threshold defined as 0.0001/*N* (*N* being the number of tested SNPs). 

The *F*_ST_ case-control analysis was performed using the –fst functionality in PLINK 1.9 [11], by comparing single markers between the PW and PB. Relevant *F*_ST_ differences were defined considering the SNPs falling in and above the 99.98th percentile distribution [3,4].

A Manhattan plot of the results was generated using the R package *qqman* [13]. *p*- and *F*_ST_ values of each SNP were plotted as a function of its position along each autosomal chromosome. The overlapping genomic regions identified by both approaches were further explored to identify linked candidate genes using the Genome Data Viewer (https://www.ncbi.nlm.nih.gov/genome/gdv/browser/genome/?id=GCF_000002315.4) developed by NCBI. To investigate the biological functions and the phenotypes that are known to be regulated by each annotated gene, we conducted a comprehensive literature search, including information from other species. Pair-wise Linkage Disequilibrium (LD) was estimated as the genotype correlation coefficient (r^2^) [14]. For all pairs of autosomal SNPs, r^2^ measures were obtained using the–r^2^–ld-window 99999–ld-window-r^2^ 0 command in PLINK v1.9 [11]. LD values were grouped into bins based on the base-pair distance between SNPs from the physical map. The average per-bin LD as a function of the base-pair distance was then used to estimate LD decay.

## 3. Results

After quality control (see above), the final number of SNPs retained for the analysis was 283,893 and no animal was discarded due to poor quality genotyping.

The GWAS analysis revealed a total of 80 highly significant Bonferroni corrected SNPs (*p* < 0.0001 (−log10 (*p*) = 9.45) located on eight autosomes (Appendix A). The corresponding Manhattan plot is reported in Figure 2a. The chicken chromosome (GGA) 01 showed the largest number of significant markers (55), and except for one marker, all the SNPs on this chromosome were located inside a 3,57 Mb region (184,995,531–188,565,711 bp) (Appendix A). Moreover, these markers on GGA01 are plotted in two single points on the Manhattan plot because they are adjacent to each other and had the same *p*-value (Figure 2a).

To further support results from GWAS, a genome-wide *F*_ST_ case-control analysis was also performed. The analysis showed a total of 66 SNPs above the selected threshold (*F_ST_* = 0.74), located on six different autosomes (GGA 01, 03,08,12,14 and 21) (Appendix A; Figure 2b). In agreement to the results provided by GWAS, the highest number of significant markers are mapped on GGA01 (52).

Combining the results from GWAS and *F*_ST_, we identified a total of 40 significant markers distributed on GGA 01, 03, 08, 12 and 21 (Table 1).

Levels of pairwise LD decreased with increasing genomic distance between SNPs (Figure 3). The Polverara breed showed moderate LD decay, with the average r^2^ falling below 0.20 after 50 Kb.

Several SNPs were adjacent or near to each other. We searched candidate genes within 250 kb-long regions (125 kb upstream and 125 kb downstream) around peak SNPs, which corresponded to median r^2^ ≥ 0.16. A total of 17 known genes were identified (Table 1).

## 4. Discussion

Potentially, there is much unrecognized beneficial genetic variation in local autochthonous animal breeds and populations [15]. As visual characteristics of animals, pigmentation traits are often used for breed identification, and represent an important phenotype of interest for breeding and research [16]. Several genome-wide studies for coat color have been conducted in livestock species including cattle [16,17,18], sheep [2,19], goat [20,21]. In this work, genome-wide analyses have been performed in the Polverara chicken breed, PW and PB subpopulations. Considering its phenotypic variability (black vs. white), this local breed has been used as a model for investigating the genetic bases of plumage color.

Based on the greater power of analysis in limiting the number of false positive signals when more than one methodology is adopted in parallel, two different approaches (GWAS and *F*_ST_) have been used in this study [3,4]. From results, the major overlap in genomic regions associated to the phenotypic differences was found on GGA01 (Table 1). The most striking result refers to a relatively narrow 0.88 Mb interval (187,660,456–188,540,546 bp). This region showed strong divergence between PW and PB. One of the most significant markers (SNP *AX-75379450*) was located within the *TYR* gene, while a total of 10 significant SNPs were concentrated in a very small interval of 0.27 Mb within the *GRM5* gene. *TYR* codes for a key enzyme in melanin biosynthesis and it has been accepted as a major gene involved in plumage color in chickens [22,23,24]. A previous study reported that *TYR* showed the greatest level of differential expression in the skin of black versus white chickens [24]. In humans [25,26,27] and mice [28], several genome-wide studies have also shown signals of association for skin or coat color in the genomic regions encompassing the *GRM5* and *TYR* genes. On GGA1, there were three other significant markers close to the *RAB38* gene whose products is a Ras-related protein. Ras-related proteins are critical regulators of cellular membrane trafficking [29] and are involved in a variety of processes, including skin pigmentation [30]. It has been reported that the mouse *RAB38* gene acts in a functionally redundant way in regulating skin melanocyte pigmentation and controls the post-Golgi trafficking of tyrosinase (*TYR*) and tyrosinase-related protein 1 (*TYRP1*) [31]. Moreover, a GWAS for chicken plumage pigmentation reported a gene belonging to the RAS family, *RAS4A*, located in the region of a significantly associated SNP [10]. On GGA08, two SNPs (*AX-77109855* and *AX-77109898*) were both located within the *NOTCH2* gene. A recent study [32] reported that Notch signaling is involved in the regulation of melanocyte development during adulthood, and *NOTCH2* contributes to the regulation of melanocyte homeostasis. Furthermore, *NOTCH2* cooperates with *c*-kit signaling during embryogenesis, and they cooperate to regulate melanocyte homeostasis after birth [32]. Therefore, in addition to the well-known *TYR* gene, it can be hypothesized that variants of the *GRM5*, *RAB38* and *NOTCH2* genes could be related to plumage color in chicken. A previous GWAS for plumage color [33], using a low-density array, revealed a significant association with SNPs mapped on the *AKT3*, *KRT7*, *PAP2* and *DDX6* genes. Yang et al., [10], in a GWAS using black and no-black chickens, showed a strong association with SNPs within *SHH* and *NUAK* genes, while Johansson and Nelson [34] reported that the *EDN3* gene is associated with dark pigmentation in two local chickens breeds. The authors did not observe any association with the candidate genes here reported. A possible reason for the lack of correspondence among studies may be the different breeds used in the comparison (and their plumage color), the array density and the statistical approaches. In this study, we have reported as candidate loci for chicken plumage color the genomic regions obtained combining the results from two different approaches applied to PW and PB. Despite the phenotypic differences, the two populations share a common genetic background [5,6,7]. This leads to minimize the confounding effects due to genetic divergence and population structure [15,20]. Moreover, some candidate genes identified here, such *TYR*, are consistent with results reported from previous studies on chicken plumage color. All of the above is likely to have helped us circumvent potential biases linked to false positive signals: the identified genes should therefore be considered rather robust results, which can contribute to explain the genetic contribution to phenotypic differences between PW and PB.

The most obvious phenotypic difference between PW and PB is the plumage color, and a number of genes involved in the determination of this phenotype have been detected in this study. It should also be pointed out that other known genes have been identified by combining GWAS and *F*_ST_. Significant markers on GGA01 were close to candidate genes involved in feed conversion ratio (*NOX4*) [35] and feed efficiency (*TMEM135*) [36] in chickens. On GGA12, the analyses revealed three markers close to *KLF15*, a gene associated with chicken growth and carcass traits [37]. It is likely that the two populations differ for additional less obvious phenotypes, such as reproductive performance or feed efficiency.

## 5. Conclusions

In poultry, plumage color is an important qualitative trait that can serve as marker useful for breed identification. Although the chicken genome is well studied, not all the genes affecting plumage color are described. Based on previous studies in other species, the present work has revealed new potential candidate genes involved in the phenotypic variability of color in local chicken populations. These results contribute insights into the genetic basis for plumage color in poultry, and confirm the great complexity of the mechanisms that control this trait. Additional research will be necessary to refine the presented results and further investigate the molecular mechanism underpinning plumage color.

## Figures and Tables

**Figure 1 animals-10-00493-f001:**
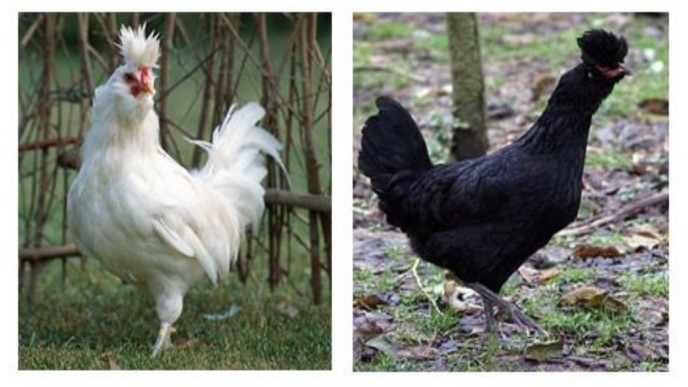
Specimens of Polverara White (PW) and Polverara Black (PB) chickens.

**Figure 2 animals-10-00493-f002:**
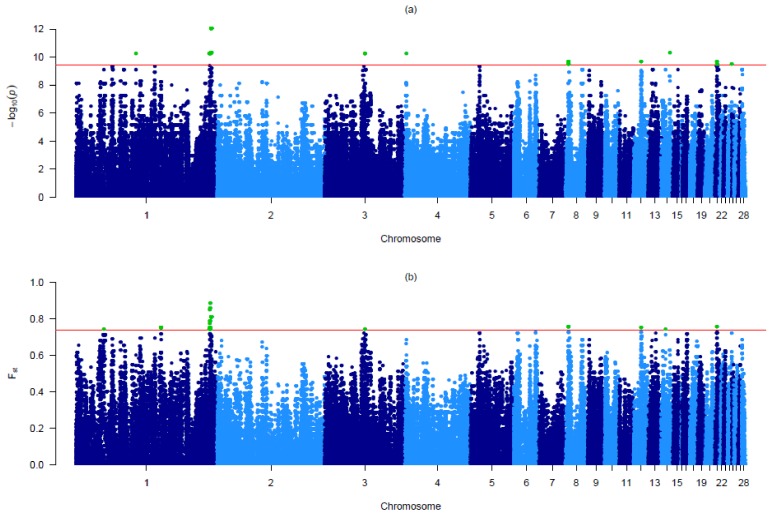
(**a**) Manhattan plot of the *p*-values in the genome-wide association study (GWAS). The horizontal lines represent the Bonferroni-corrected genome-wide significance (red; *p* < 0.0001); (**b**) Manhattan plot of the genome-wide fixation index (*F*_ST_). The horizontal line represents the genome-wide significance single nucleotide polymorphisms (SNP) above the 99.98th percentile distribution) (*F*_ST_ = 0.74). Significant SNPs are highlighted in green.

**Figure 3 animals-10-00493-f003:**
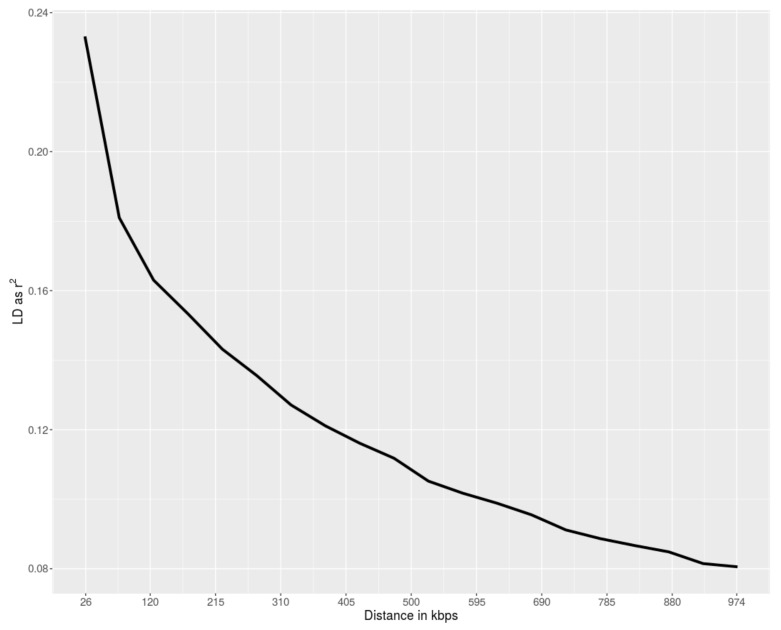
Linkage Disequilibrium decay (measured as r^2^) as a function of inter-marker distance (Kbp) in the Polverara breed.

**Table 1 animals-10-00493-t001:** Overlapping significant markers identified by GWAS and *F*_ST_ and associated genes.

					**Nearest Gene**	
GGA	SNP	Position (bp)	*p*-Value	*F* _ST_	Name	Distance (kb)
1	AX-75371751	184995531	5.45e-11	0.745	*MAML2*	2.01
1	AX-75373909	185836576	5.45e-11	0.773	*LOC107052349*	2.90
1	AX-75374539	186058014	5.45e-11	0.858	*CCDC67*	29.43
1	AX-75375587	186464423	5.45e-11	0.858	*FAT3*	Within
1	AX-75376255	186722445	5.45e-11	0.886		
1	AX-75376262	186735600	5.45e-11	0.886		
1	AX-75378645	187660456	9.01e-13	0.809	*NAALAD2*	Within
1	AX-75378836	187723578	9.01e-13	0.809	*FOLH1*	2.90
1	AX-75378888	187743605	9.01e-13	0.809	*FOLH1*	22.92
1	AX-75379333	187911192	9.01e-13	0.809	*NOX4*	8.02
1	AX-75379334	187911433	9.01e-13	0.809	*NOX4*	8.26
1	AX-75379450	187960805	9.01e-13	0.809	*TYR*	Within
1	AX-77278759	188025840	9.01e-13	0.809	*GRM5*	Within
1	AX-75379693	188066880	9.01e-13	0.809		
1	AX-75379724	188079273	9.01e-13	0.809		
1	AX-75379753	188089989	9.01e-13	0.809		
1	AX-75379761	188093458	9.01e-13	0.809		
1	AX-75379775	188096972	9.01e-13	0.809		
1	AX-75379792	188102761	9.01e-13	0.809		
1	AX-75379800	188106002	9.01e-13	0.809		
1	AX-75379813	188112765	9.01e-13	0.809		
1	AX-75380172	188238879	9.01e-13	0.809		
1	AX-75380766	188476552	9.01e-13	0.809	*RAB38*	88.89
1	AX-75380808	188490865	9.01e-13	0.809	*RAB38*	103.21
1	AX-80852333	188493037	9.01e-13	0.809	*RAB38*	105.38
1	AX-75380927	188538625	9.01e-13	0.809	*TMEM135*	61.51
1	AX-75380931	188540546	9.01e-13	0.809	*TMEM135*	59.59
3	AX-76506116	55929533	5.45e-11	0.745	*HBS1L*	9.40
3	AX-76506117	55930178	5.45e-11	0.745	*HBS1L*	9.40
8	AX-77109355	4012906	2e-10	0.757	*CRIP1*	7.01
8	AX-77109358	4014014	2e-10	0.757	*CRIP1*	8.12
8	AX-77109696	4164384	2e-10	0.757	*SEC22B*	Within
8	AX-77109700	4167984	2e-10	0.757		
8	AX-77109855	4230320	2e-10	0.757	*NOTCH2*	Within
8	AX-77109898	4249450	2e-10	0.757		
12	AX-75680106	10597665	2e-10	0.755	*KLF15*	37.98
12	AX-75680164	10627473	2e-10	0.755	*KLF15*	8.17
12	AX-75680170	10629579	2e-10	0.755	*KLF15*	6.07
21	AX-76239008	2640299	2e-10	0.757	*C21H1ORF159*	0.53
21	AX-76239099	2657895	2e-10	0.757	*C21H1ORF159*	1.85

Note: Gallus gallus chromosome number, GGA; single nucleotide polymorphism, SNP.

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
