# Peer review of "Genome-Wide Analyses Identifies Known and New Markers Responsible of Chicken Plumage Color"

_animals, 2020, doi:10.3390/ani10030493_

Round 1

Reviewer 1 Report

The manuscript “Genome-wide Analyses Identifies Known and New Markers Responsible of Chicken Plumage Color” by Mastrangelo and co-authors explores different genomic regions involved in the pigmentation of chickens plumage of Polverara chicken breed. By the mean of combined genome-wide approach (GWAS and Fst), a total of 40 significant markers were identified involved in this trait. It is evident that the manuscript is well written and presented. Few minor points should be considered and addressed by the authors prior to being submitted for publication as detailed below:

- Please mention in the abstract and the introduction what is GWAS. As described in the M&M it should be Genome-wide association study.

-FST should be clarified as (fixation index, a genetic measurement of population differential)

-I think the authors have obtained an ethical permission to keep birds and take blood sample, if yes, then this should be provided.  

-Line 52: The authors may provide the names of the six involved chicken breeds.

-Figure 2: what do green, blue, and dark blue colours refer to?

- Could the authors provide more information about the autosomes refine criteria from table S1 to S2 to table 1.

-Line 187-189: “Determination of these genomic regions with candidate genes will also help to protect genetic resources, such as the Polverara populations.” How could this protect the genetic resources, please clarify!

Reviewer 2 Report

In this paper the authors report results of a GWAS and Fst outlier approach aimed at identifying the genomic basis for plumage color variation in a dichromatic chicken breed. The analysis identified several SNPs associated with genes known to be involved in melanogenesis, along with several other genes that have not been previously implicated in chickens.

While I appreciate the motivations of identifying genetic resources in ‘local’ breeds, I’m afraid the paper did not provide a compelling argument for the scientific contribution of this study. First, I found the methods to be conspicuously lacking in detail, to a degree that made it difficult to assess the validity of the results. Likewise, the writing suffered from lack of clarity due to numerous grammatical errors and awkward phrasing. Finally, and perhaps critically, I found the authors’ assertion that “very few studies on a genome-wide scale have been performed to identify the molecular bases for pigmentation in chickens plumage” (lines 63-64) to be grossly inaccurate, and ignores a significant body of literature that address this exact topic. As a consequence, it is difficult to evaluate the precise contribution of this study in the context of the present state of the field.

Line 24: ‘chickens plumage’ grammatically incorrect.

Line 25: change ‘color plumage’ to ‘plumage color’

Lines 34-36: unclear statement

Lines 63-64: I disagree with this assessment—there is an extensive literature of genetics of plumage coloration in chickens.

Line 70: Please provide more information on the animals selected for the study. Were they known to be unrelated?

Lines 86-87: “The top 0.9998 SNPs of the percentile distribution”  Unclearly worded, I suggest rephrasing.

Lines 118-119: What is the justification for the use of the 250 kbp region?

Line 124: “…which supposes…” is grammatically incorrect.

Lines 132-134: In my opinion, there are not ‘two different approaches’ utilized here, but rather two metrics calculated from the same SNP genotype dataset. This is different from performing two distinct experiments to corroborate the results.

Lines 183-184: Considering the immense variety in chicken plumage coloration, why should we assume there are only a limited number genes involved?

Reviewer 3 Report

The study described a GWAS approach to analyze loci in plumage color in the Polverara chicken breed. In total, 37 animals are used and genotyped by Affymetrix 600K array. The analysis yielded 40 significant markers, and the authors discuss the reevance of nearby genes, some of them know previously to affect plumage colot (TYR).

The population history of the Polverara breed should be introduced in more detail - are the black and white populations separated or breeding together? Are there multiple color variants and is there any indication from this how many genes might be involved?

Minor points:

Line 19: "potential candidate genes"... it is enough to state "candidate genes"

Line 174: feed conversion ratio
